# Influence of Additional Bracing Arms as Reinforcement Members in Wooden Timber Cross-Arms on Their Long-Term Creep Responses and Properties

**Muhammad Rizal Muhammad Asyraf** [1,*] , **Mohamad Ridzwan Ishak** [1,2,3,*], **Salit Mohd Sapuan** [3,4] **and Noorfaizal Yidris** [1]

1 Department of Aerospace Engineering, Universiti Putra Malaysia, UPM, Serdang 43400, Selangor, Malaysia; nyidris@upm.edu.my
2 Aerospace Malaysia Research Centre (AMRC), Universiti Putra Malaysia, UPM, Serdang 43400, Selangor, Malaysia
3 Laboratory of Biocomposite Technology, Institute of Tropical Forestry and Forest Products (INTROP), Universiti Putra Malaysia, UPM, Serdang 43400, Selangor, Malaysia; sapuan@upm.edu.my
4 Advanced Engineering Materials and Composites Research Centre (AEMC), Department of Mechanical and Manufacturing Engineering, Universiti Putra Malaysia, UPM, Serdang 43400, Selangor, Malaysia
* Correspondence: asyrafriz96@gmail.com (M.R.M.A.); mohdridzwan@upm.edu.my (M.R.I.)

**Abstract:** Previously, numerous creep studies on wood materials have been conducted in various coupon-scale tests. None had conducted research on creep properties of full-scale wooden cross-arms under actual environment and working load conditions. Hence, this research established findings on effect of braced arms on the creep behaviors of Virgin Balau (*Shorea dipterocarpaceae*) wood timber cross-arm in 132 kV latticed tower. In this research, creep properties of the main members of both current and braced wooden cross-arm designs were evaluated under actual working load conditions at 1000 h. The wooden cross-arm was assembled on a custom-made creep test rig at an outdoor area to simulate its long-term mechanical behaviours under actual environment of tropical climate conditions. Further creep numerical analyses were also performed by using Findley and Burger models in order to elaborate the transient creep, elastic and viscoelastic moduli of both wooden cross-arm configurations. The findings display that the reinforcement of braced arms in cross-arm structure significantly reduced its creep strain. The inclusion of bracing system in cross-arm structure enhanced transient creep and stress independent material exponent of the wooden structure. The addition of braced arms also improved elastic and viscoelastic moduli of wooden cross-arm structure. Thus, the outcomes suggested that the installation of bracing system in wooden cross-arm could extend the structure's service life. Subsequently, this effort would ease maintenance and reduce cost for long-term applications in transmission towers.

**Keywords:** balau wood; cross-arm; transmission tower; bracing system; creep; findley's power law model; burger model

## 1. Introduction

Anisotropic materials has been widely used in many large structures such as bridges, buildings, pedestrian walkways and cross-arm in transmission tower. Wood and wood composites are those common anisotropic materials used to build many civil structures [1–4]. However, many studies reported that most wooden structures experienced premature failures after long period of service, especially when continuously exposed toward extreme weather [5–7]. These premature failures are also contributed by creep and natural defect such as wood ageing process, which subsequently may lead to structural collapse [8–13].

Creep is one of the major concerned by structural engineers in order to eliminate any possibilities of structural collapse during its service operation. Creep is a term referred to the tendency of a solid material to move slowly or deform permanently under the

influence of persistent mechanical stresses. They are divided into several phases starting from instantaneous deformation followed by primary (transient), secondary (steady-state) and tertiary (accelerated). This mechanical phenomenon usually due to shear yielding, chain slippage, void formation, and also breakage of fibres [14,15]. To be specific, the creep response of wooden materials is dependent based on their level of stress, operation time, and surrounding temperature [16]. Numerous creep studies of wooden materials have been carried out by many researchers in many ways including development of creep test rigs [17–19], and creep numerical analyses [20,21] and coupon tests [22,23].

Currently, creep properties of wooden cross-arm in latticed transmission tower is still unexplored based on the previous literatures. Earlier, several research studies reported that the wooden cross-arm has shorter life span (less than 20 years of service) than its life expectancy due to its natural wood defects [24,25]. The wood defects occur due to natural fibre and wood defects as the wood is exposed to a constant loading for a prolonged time [26–28]. Hence, this issue has led engineers and researchers to find a solution in order to extend lifespan of the wooden cross-arms. One of the solution is the implementation of bracing system in the current cross-arm structure as suggested by Sharaf et al. [29]. Thus, this study evaluates the influence bracing system on long-term mechanical performance of wooden cross-arm.

Evaluating creep properties and responses of a full-scale wooden cross-arm used in 132 kV transmission tower could eliminate numerous exaggerated factors that happen in coupon scale test. The geometry and profile of the material could be neglected when the test is conducted in coupon scale, such as flexural, tensile, and compressive properties. Thus, more reliable data collection would be achieved when a full-scale size cross-arm is used in carrying out creep test to understand the mechanical behaviour during long-term loading condition. The investigation of long-term behaviour of main component member for the cross-arm structure could provide a more holistic perspective in order to evaluate the behaviour of the whole structure either with or without bracing.

Nowadays, many latticed transmission towers are still used the conventional wooden cross-arm to transmit electrical power. A literature survey revealed that no previous works have evaluated the creep properties of full-scale wooden cross-arms used for 132 kV towers [30,31]. Thus, this manuscript is expected to examine the effect of addition braced arms on the creep properties and responses of wooden cross-arms with its actual loading conditions. The study also intended to set a baseline for creep profiling of full-scale wooden cross-arms. At the end, the outputs from this study would create a practical perspective for engineers to understand the long-term mechanical performance of the conventional wooden structure.

## 2. Methodology

### 2.1. Materials

Balau timber wood or *Shorea Dipterocarpaceae* was used as a cross-arm's material to examine long-term creep behaviours. The cross-arm in 132 kV latticed transmission tower was composed of one tie member and two main members. All cross-arm members were fabricated from the same Balau tree trunk obtained at Bahau, Malaysia. The timber wood was cut individually to form continuous square section with the dimension of 102 mm × 102 mm. In term of size, the lengths of the main and tie members of the cross-arm was 3651 mm and 3472 mm, respectively. Each of the cross-arm members was joined together by using both bolts and nuts, as well as its mild steel fastener brackets. These members were assembled using fastener brackets in order to be incorporated on the creep test rig with constant loading.

For the braced arms, they connected with main and tie members by using custom-made fastener brackets. The braced arms were in square section beams with dimension of 50 mm × 50 mm. The arms are also made from Balau timber wood. In total, there are five braced arms interconnected with main and tie members. Those braced arms encompassed of two long members (connected in the middle of the tie member to the end of the main

member) and three short members (joined at the middle to the middle of every member). The lengths of the long (tie-main), short (tie-main), and short (main-main) bracing members were 186, 50, and 40 cm, respectively.

### 2.2. Methods

The test was set up based on actual position of cross-arm in latticed transmission tower. The creep test was performed on the specialised cantilever beam creep test rig used for cross-arm testing. To evaluate the strain measurement, ten dial gauges were positioned 0.61 m in between five points under two main members. Figure 1 illustrates the positions and the length of between the dial gauges under the wooden cross-arm. The load was implemented at the bottom side of the joining parts of all members.

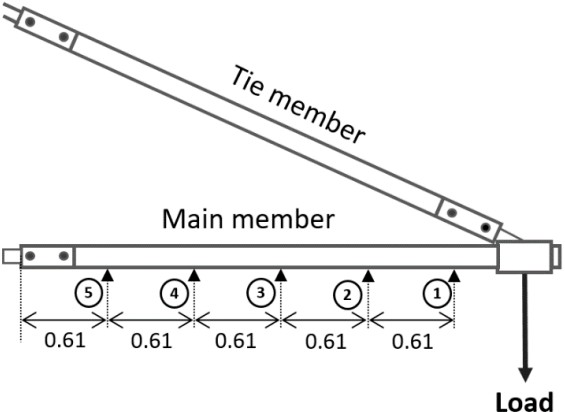

**Figure 1.** Positions of dial gauges under the cross-arm to measure creep strain pattern, in meter.

Detail drawing with dimensions of cantilever beam creep test rig with wooden cross-arm is displayed in Figure 2. The test rig was manufactured from mild steel square hollow section with dimension of 100 mm × 100 mm and 1.9 mm thickness. The rigidity and specifications of the test rig can be found in Table 1 [19]. The wooden cross-arm was installed on the test rig using forklift and it is manually fixed on the test rig's fastener brackets. A dead load was assembled at free end of the cross-arm to mimic the actual cross-arm conditions in the transmission tower as shown in Figure 3. In general practice, the wooden cross-arm is installed in suspension latticed transmission tower to carry the electric cables and insulators [32]. To be specific, the cross-arm was mounted on the test rig at height of 2100 mm from ground to hang the dead load for the creep test.

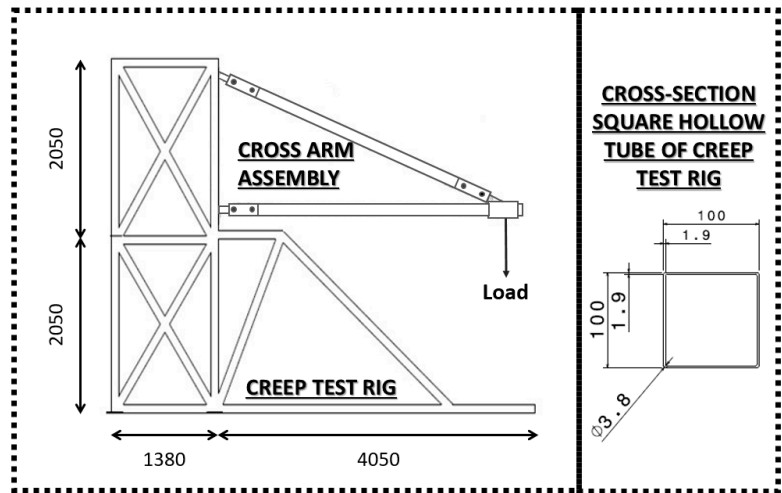

**Figure 2.** Schematic diagram of wooden cross-arm used in creep test rig.

**Table 1.** Specification of material of creep test rig [19].

| Properties | Specification |
| --- | --- |
| Material | Mild steel |
| Tensile strength, MPa | 766 |
| Yield strength, MPa | 572 |
| Pipe shape | Square hollow section |
| Pipe size (width/height/thickness), mm | 100/100/1.9 |
| Total size (width/length/height), mm | 1525/430/4100 |

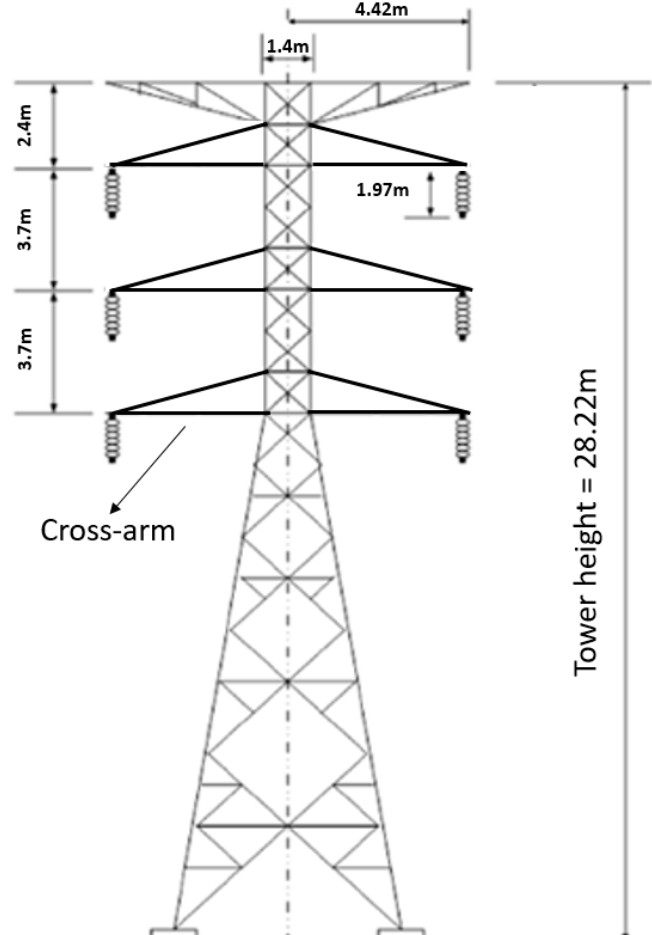

**Figure 3.** Dimension of latticed transmission tower. Adapted with permission from IEEE, 2021 [33].

The geometry and dimensions of latticed transmission tower is shown in Figure 3 [33]. The steel tower and wooden cross-arm was connected using specialised fittings. The assemblymen of wooden cross-arm to steel tower is manually fixed by bolt and nut. Based on Figure 4, the cross-arm was attached to the steel tower by sandwiching two steel plate fittings with end cross-arm member and they are manually connected by using bolt and nut.

As shown in Figure 5, the bracing members were connected with cross-arm's members via custom-made fastener brackets as aforementioned. The custom-made steel fastener bracket was designed and manufactured specifically to fit the shape of cross-arm. For the assembly process, the brackets were manually installed by using bolt and nut after the cross-arm was completely placed and fixed at the test rig. Figure 6 displays schematic diagram of two set configurations of cross-arm structure, which are current design (without braced arms) and braced design (with braced arms) configuration.

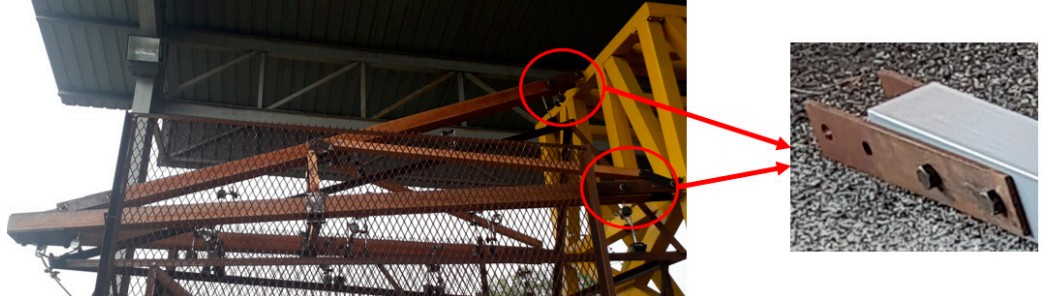

**Figure 4.** Connection of cross-arm to steel tower using steel plate fittings.

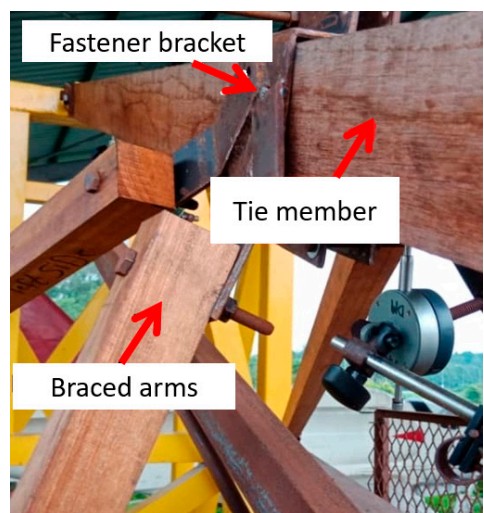

**Figure 5.** Mild steel fastener bracket joins the braced arms with tie members.

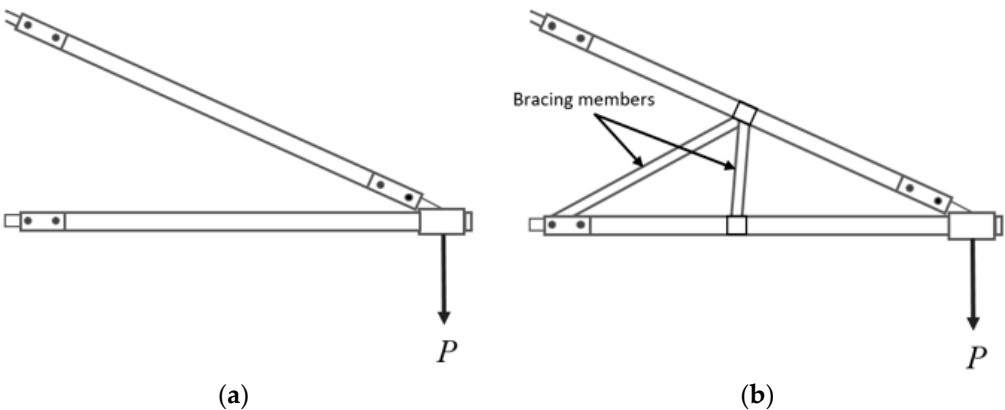

**Figure 6.** Cross-arm configurations: (**a**) current design configuration; and (**b**) braced design configuration.

In term of dead load, a hanging load was weighed based on actual working load of cross-arm (6.347 kN) before the experiment started. The test fulfilled the requirement time as in ASTM D2990, which evaluated the creep test at 1000 h of operation. Specifically, readings were taken at several specific time periods (0.1, 0.2, 0.5, 1, 2, 5, 20, 50, 100, 200, 500, 700 and 1000 h) to observe creep deformation. The condition of the experiment was set at open area that constantly exposed to actual tropical weather. At the end of experiment, the comparison of current (without bracing members) and braced (with bracing members) cross-arm designs was carried out in terms of long-term creep properties.

2.2.1. Creep Properties of Cross-Arms

A wooden cantilever beam structure usually experiences viscoelastic behaviour when a constant loading is continuously applied at the end of the beam. In common practice, the beam exhibits constant tension and compression actions on opposite sides of the beam, which induces a series of strain pattern depending on the applied load. When the displacement remains at certain positions along the time period, the viscoelastic beam usually expresses the stress response on the beam and gradually decreases. This shows that the viscoelastic beam responds to the material's viscous characteristic, which would decrease the total stress [34,35]. Similarly, a beam that is exposed to a constant load continues to deform as the material relaxes.

In general, the static elastic modulus ($E_e$) of the beam as shown in Figure 7 can generated based on Equation (1):

$$E_e = \frac{4PL^3}{ybh^3} \tag{1}$$

where, $y$ is the deflection at the beam (m); $E_e$ is the static elastic modulus (N/m$^2$); $P$ is the force exerted on the beam (N); $L$ is the total length (m) and $b$ and $h$ are the width and thickness of the beam (m), respectively.

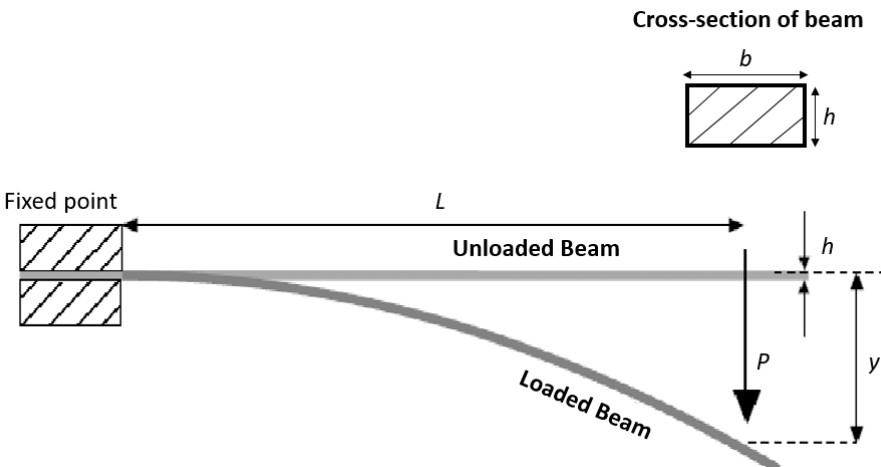

**Figure 7.** Schematic diagram of cross-arm structure when exposed applied force at the end of the cross-arm structure.

Given that the y deflection is already known, the maximum bending stress can be predicted by using Equation (1) [36]. In general, the maximum stress experienced by the cross-arm is usually located at fixed point $x = 0$, whereas the minimum stress is exhibited at the loading end $x = L$. The maximum and minimum stresses of the beam are formulated on the basis of Equation (2).

$$\sigma = \frac{P(L-x)\frac{h}{2}}{I} = \frac{6P(L-x)}{bh^3} \tag{2}$$

Equation (3), are formulated based on Hooke's law equation. The Equation (3) is functioned to calculate the creep strain at a specific time and specific location across the beam.

$$\varepsilon_t = \frac{\sigma_n}{E_e} \tag{3}$$

where, $\varepsilon_t$ is the creep strain at a specific time and location point across the beam. $\sigma_n$ is specific stress and $E_e$ is the static elastic modulus at the specific point on the cross-arm.

### 2.2.2. Constitutive Creep Models

Findley power law model is an empirical mathematical model that simulates the creep properties of anisotropic material. The model is presented as in Equation (4) [37].

$$\varepsilon_t = At^n + \varepsilon_0 \tag{4}$$

where, $A$ and $n$ as transient creep strain and time exponent respectively, while $\varepsilon_0$ is the instantaneous strain after exerted the load.

To assess the time-dependent responses of the Balau wooden material on the basis of the flexural information, a reliable creep model has to be recognised. One of the models used in order to identify the relationship between the structure and creep behaviour was Burger model [5,34]. This model can be expressed in Equation (5).

$$\varepsilon_t = \varepsilon_e + \varepsilon_d + \varepsilon_v \tag{5}$$

The mathematical formulation in Equation (5) comprises $\varepsilon_e$, $\varepsilon_d$ and $\varepsilon_d$, which are called the elastic strain, viscoelastic strain and viscous strain, respectively.

Equation (6) was derived on the basis of Equation (5) and the physical elements of Burger models, such as spring and dashpot elements.

$$\varepsilon_t = \frac{\sigma}{E_e} + \frac{\sigma}{E_d}[1 - exp(-t/\tau)] + \frac{\sigma}{\eta_v}t; \ \tau = \frac{E_d}{\eta_d} \tag{6}$$

At this point of view, these three strain components later were derived into stress, elastic modulus and viscoelastic modulus as shown Equation (6). Both elastic and viscoelastic moduli are essential behaviour of the material. Perez et al. [38] and Chandra and Sobral [39] established that the Burger model compromises of combination of three elements including a linear elastic spring, a dash pot, and a Kelvin-Voight element (dash-pot and combination of dash-pot and spring). Figure 8 visualise the long term behaviour of viscoelastic material under Burger model.

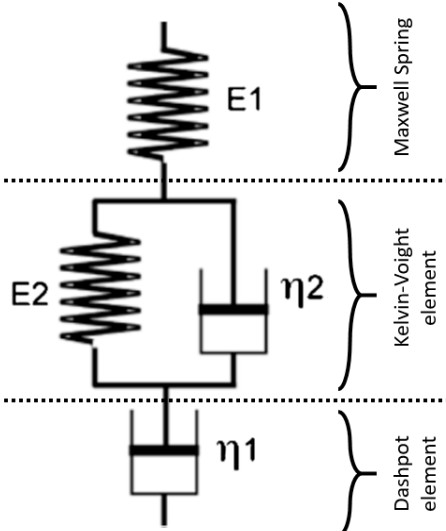

**Figure 8.** Schematic diagram of Burger model.

## 3. Results and Discussion

### 3.1. Strain-Time Curve

The creep strain-time graphs for both current and braced wooden cross-arm configurations at each point of main members are presented in Figure 9. Based on the curves, it can be seen that creep pattern are divided into three phases which are instantaneous deformation,

primary and secondary creeps. As expected in the early experiment, the highest creep strain is located at y3, which at the centre of cross-arm's main member beams. This was probably due to the each of main members of the structure experienced compression from both ends when the force is exerted at free end of the cross-arm. As mentioned by Kanyilmaz (2017) [40], a simple interconnected members of a structure without bracing systems would induce inelastic behaviour, which tends to experience buckling at the centre of the structure arm. This established that the operational cross-arm would experience buckling action due to external forces from the dead weigh at the end of the structure.

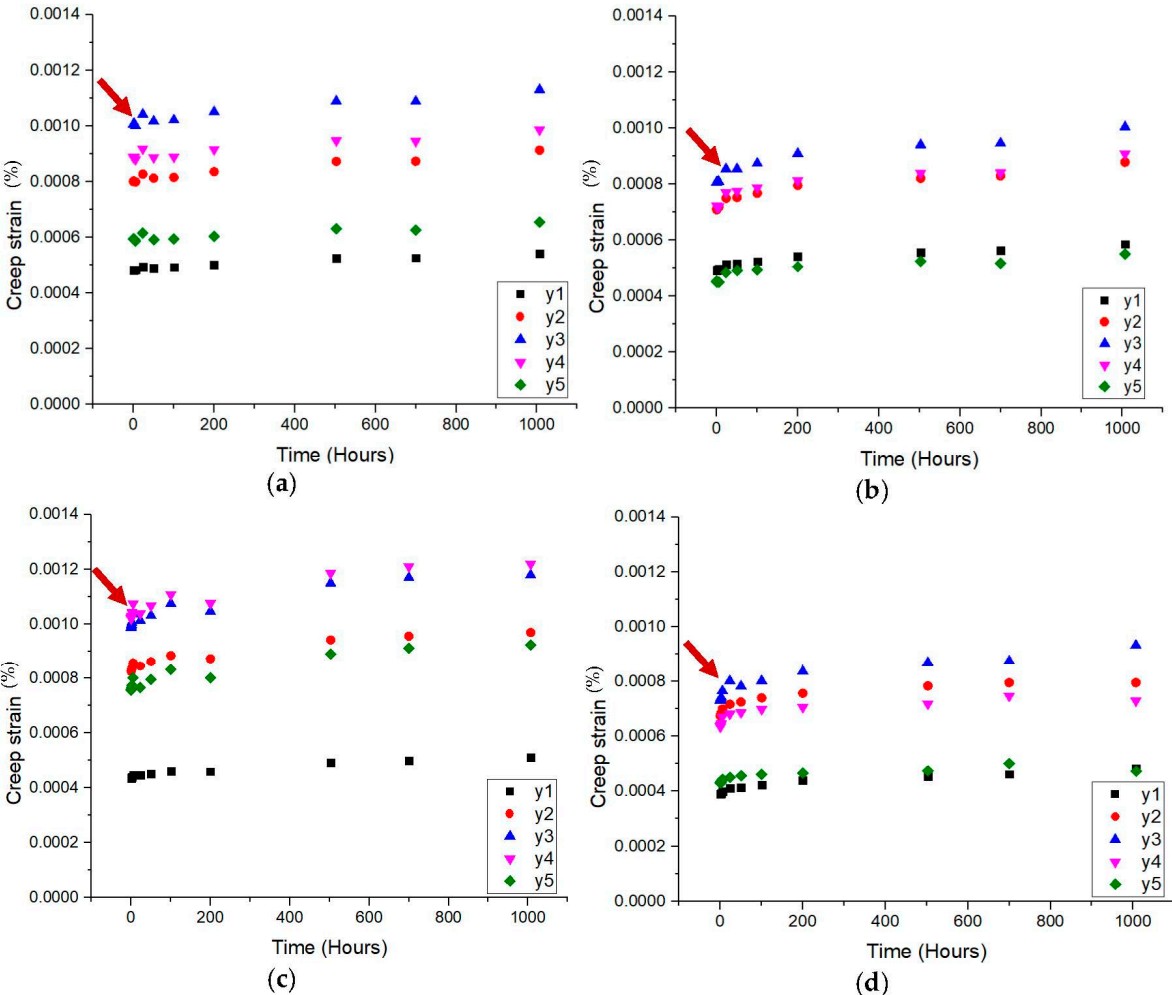

**Figure 9.** Creep strain-time curves for current wooden timber cross-arm for (**a**) left and (**c**) right; braced wooden timber cross-arm for left (**b**) and right (**d**) main member.

In the meantime, the findings also depicts the creep strain patterns have two distinct stages: elastic and viscoelastic stages. However, the transition period from the elastic period to the viscoelastic phase was extended for the current design cross-arm (red arrow's location shown in Figure 9). This established that the braced wooden cross-arm provide the structure become more stable in viscoelastic stage.

Based on Figure 9b,d, the curves in both left and right main members of braced design cross-arm exhibit a similar pattern. The similar creep strain pattern for both left and right wooden cross-arm is probably due to the addition of bracing system provide better structural integrity. Subsequently, a symmetrical shape and deformation pattern would permit during the cross-arm service to grasp the power cables and insulators, which would reduce any potential sudden failure of the structure after years of service.

Besides that, the installation of the brace arms reduces the creep strain along the operation time. This can be observed where the instantaneous deformation of the braced design cross-arm are noticeably less than current design cross-arm at any points along the individual member. For instance, at the middle of cross-arm (y3), the finding displays that the creep strain value for the current wooden cross-arm was higher than the braced wooden cross-arm. At this point of view, the inclusion of bracing system in the structure would significantly provide higher creep resistant performance by reduced the creep strain approximately 31%. This outcomes is tally with a research conducted by Patil et al. [41]. They mentioned that the improvement of mechanical response of a structure is significant enhanced as the addition of bracing members could resist lateral forces which subsequently avoid from buckling.

### 3.2. Findley Power Law Model

Table 2 tabulates the values of *A* parameter and stress-independent material exponent, *n*. *A* and *n* parameters were discovered based on Equation (4) by using Origin Pro 2016.

**Table 2.** Average parameters obtained from Findley power law for current and braced wooden cross-arms.

| Main Member Arm | Location | A | | n | | Adj. R$^2$ | |
|---|---|---|---|---|---|---|---|
| | | Current Cross-Arm | Braced Cross-Arm | Current Cross-Arm | Braced Cross-Arm | Current Cross-Arm | Braced Cross-Arm |
| Right | 1 | $5.818 \times 10^{-7}$ | $6.086 \times 10^{-6}$ | 0.669 | 0.395 | 0.973 | 0.989 |
| | 2 | $6.977 \times 10^{-7}$ | $1.005 \times 10^{-5}$ | 0.726 | 0.402 | 0.957 | 0.981 |
| | 3 | $1.128 \times 10^{-6}$ | $1.255 \times 10^{-5}$ | 0.673 | 0.396 | 0.936 | 0.977 |
| | 4 | $1.927 \times 10^{-7}$ | $1.350 \times 10^{-5}$ | 0.895 | 0.372 | 0.888 | 0.951 |
| | 5 | $4.224 \times 10^{-8}$ | $1.429 \times 10^{-5}$ | 1.045 | 0.278 | 0.833 | 0.938 |
| Left | 1 | $1.647 \times 10^{-6}$ | $6.331 \times 10^{-6}$ | 0.551 | 0.389 | 0.981 | 0.991 |
| | 2 | $5.076 \times 10^{-6}$ | $2.969 \times 10^{-5}$ | 0.486 | 0.234 | 0.958 | 0.992 |
| | 3 | $6.377 \times 10^{-6}$ | $1.952 \times 10^{-5}$ | 0.496 | 0.331 | 0.933 | 0.960 |
| | 4 | $5.313 \times 10^{-6}$ | $6.858 \times 10^{-5}$ | 0.528 | 0.123 | 0.920 | 0.958 |
| | 5 | $4.278 \times 10^{-6}$ | $2.682 \times 10^{-5}$ | 0.533 | 0.155 | 0.917 | 0.928 |

The obtained values of *A* parameter are resumed on Figure 10. From Figure 10, it is apparent that *A* parameter is increased for braced wooden cross-arm. This result acknowledged that the implementing of bracing system could enhanced the transient creep strain of the structure. On the other hand, the left main member had higher *A* parameter value compared to the right main member in both cross-arms. This indicated that the left main member had grain orientation in either radial, tangential, or longitudinal direction which differed in terms of their creep resistance performance [42,43].

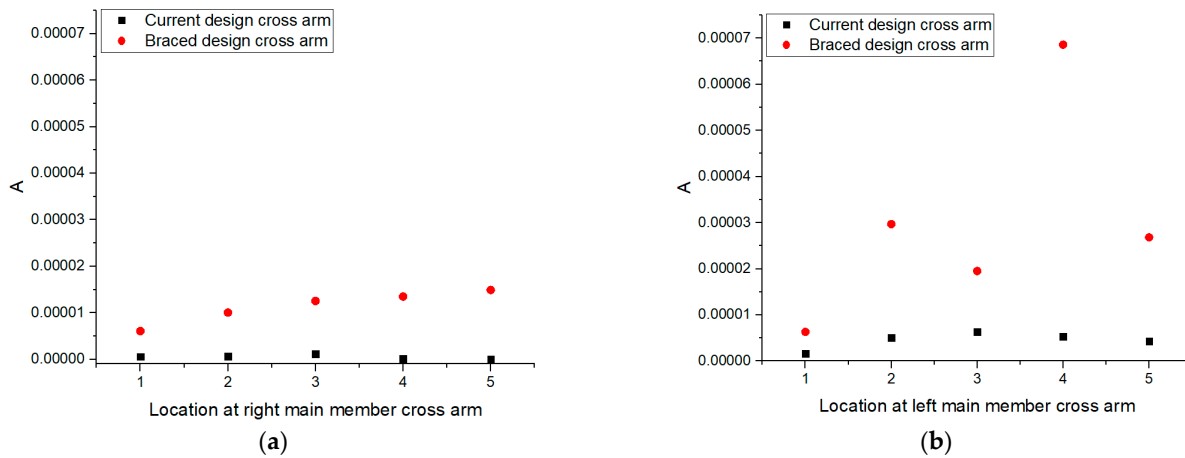

(a)  (b)

**Figure 10.** *A* parameter for current and braced wooden cross-arms: (**a**) right; (**b**) left main members

In terms of stress-independent material exponent, the findings showed that the average value for the existing right cross-arm (0.8016) was higher than the existing left cross-arm (0.5188) as shown in Table 3. This was probably due to both members of cross-arms having different grain direction and orientation, which would affect the *n* exponent. According to Hill (2006) [44], the cell wall swelled significantly more either in radial or tangential directions rather than longitudinal orientation during water absorption process. This was affected by winding angle of microfibrils within the wood fibre layer. On the other hand, the manufacturer might cut each beam separately from different tree trunks or different heights. According to Machado et al. [45] and Van Duong and Matsumura [46], a significant decrease of wood density would affect their bending properties when the height of timber varied. Apart from that, different heights of tree may contribute to different chemical compositions, such as cellulose, hemicellulose, lignin, and ash, which affect their physico-mechanical properties [47]. These factors might contribute to different values of stress-independent material exponent, *n*, for left and right main members of the same cross-arm. However, when the cross-arm was incorporated and added with bracing arms, it seemed that the right and left main members of the cross-arm had almost similar value of *n* exponent, which are 0.3686 and 0.2464, respectively (Table 3). This attribute was due to the addition of brace arms in wooden cross-arm structure which would improve the distribution of stress across the cross-arm members.

**Table 3.** Stress independent material exponent, *n* of both current and braced wooden cross-arms.

| Cross-Arm Configuration | Current | | Braced | |
|---|---|---|---|---|
| | **Right** | **Left** | **Right** | **Left** |
| Stress independent material exponent, *n* | 0.8016 | 0.5188 | 0.3686 | 0.2464 |

Lastly, the adjusted regression (Adj. $R^2$) forms for both existing designs (WOB) and braced (WB) wooden cross-arm had high value, narratively close to 1. This showed that the Findley's power law model fit the experimental data very well. Moreover, it explained that the creep of both cross-arms experienced two levels of creep, which are primary and secondary stages. This model forecasted the secondary stage well, but it cannot forecast the tertiary creep, which can determine the time of failure. On the other hand, the braced wooden cross-arm (0.989–0.928) exhibited higher Adj. $R^2$ value compared to the existing design (0.981–0.833). This explained that the braced wooden cross-arm followed the creep principles involving primary and secondary creep stages, and the creep data produced was less exaggerated due to better structural integrity.

*3.3. Burger Model*

Experimental graphs were fitted by means of the Burger model. A computational software, OriginPro 7.5, was implemented to plot a non-linear curve fit to identify four parameters ($E_e$, $E_d$, $\eta_d$, $\eta_k$) as shown in Table 4. The Burger model is usually executed in creep data evaluation due to its elaborate elastic and viscoelastic properties of anisotropic beam based on the working load within the creep period. In general, the elastic material demonstrated no residual deformation when the stress was detached from the structure. On the other hand, the viscoelastic property displayed a stress relaxation condition over time. The basis of the model was a combination of Burger's elements within the working load condition. However, the structure system would differently respond in terms of creep strain depending on the types of elements in the model [21].

Figure 11 depicts the values of $E_e$ at different locations in wooden cross-arm for both current and additional bracing arms. The elasticity parameter or $E_e$ was obtained from the data of instantaneous creep strain after execution of stress. From the results, it showed that the elastic performance of the cross-arm's beam decreased from the fixed point to free end. This happened due to stress along the cantilever beam which would decrease linearly from fixed to free end [34]. Apart from that, the finding displayed that braced cross-arm had higher $E_e$ value compared to the existing cross-arm. This could be due to the retrofitting of

braced arms which provided a sufficient restraining force to ease the residual deflection. Moreover, the increasing elastic modulus, $E_e$, indicated the enhancement of the tensile modulus [48]. This would increase their stiffness to resist plastic deformation to maintain their working load during their long-term service period [49].

**Table 4.** Average parameters obtained from Burger model for both current and braced wooden cross-arms.

| Main Member Arm | Location | $E_e$ | | $\eta_k$ | | Adj $R^2$ | |
|---|---|---|---|---|---|---|---|
| | | Current Cross-Arm | Braced Cross-Arm | Current Cross-Arm | Braced Cross-Arm | Current Cross-Arm | Braced Cross-Arm |
| Right | 1 | $4.62 \times 10^{10}$ | $4.53 \times 10^{10}$ | $4.11 \times 10^{14}$ | $3.72 \times 10^{14}$ | 0.970 | 0.991 |
| | 2 | $5.52 \times 10^{10}$ | $6.22 \times 10^{10}$ | $4.53 \times 10^{14}$ | $4.07 \times 10^{14}$ | 0.961 | 0.987 |
| | 3 | $6.54 \times 10^{10}$ | $8.20 \times 10^{10}$ | $6.15 \times 10^{14}$ | $5.37 \times 10^{14}$ | 0.938 | 0.983 |
| | 4 | $9.91 \times 10^{10}$ | $12.2 \times 10^{10}$ | $9.84 \times 10^{14}$ | $7.60 \times 10^{14}$ | 0.888 | 0.973 |
| | 5 | $18.5 \times 10^{10}$ | $24.4 \times 10^{10}$ | $19.2 \times 10^{14}$ | $20.9 \times 10^{14}$ | 0.824 | 0.969 |
| Left | 1 | $5.11 \times 10^{10}$ | $5.59 \times 10^{10}$ | $3.41 \times 10^{14}$ | $2.38 \times 10^{14}$ | 0.973 | 0.851 |
| | 2 | $5.33 \times 10^{10}$ | $6.35 \times 10^{10}$ | $3.67 \times 10^{14}$ | $3.43 \times 10^{14}$ | 0.941 | 0.695 |
| | 3 | $6.70 \times 10^{10}$ | $8.74 \times 10^{10}$ | $3.86 \times 10^{14}$ | $3.50 \times 10^{14}$ | 0.922 | 0.802 |
| | 4 | $8.53 \times 10^{10}$ | $13.4 \times 10^{10}$ | $5.51 \times 10^{14}$ | $8.74 \times 10^{14}$ | 0.901 | 0.559 |
| | 5 | $14.5 \times 10^{10}$ | $24.9 \times 10^{10}$ | $7.26 \times 10^{14}$ | $19.9 \times 10^{14}$ | 0.910 | 0.518 |

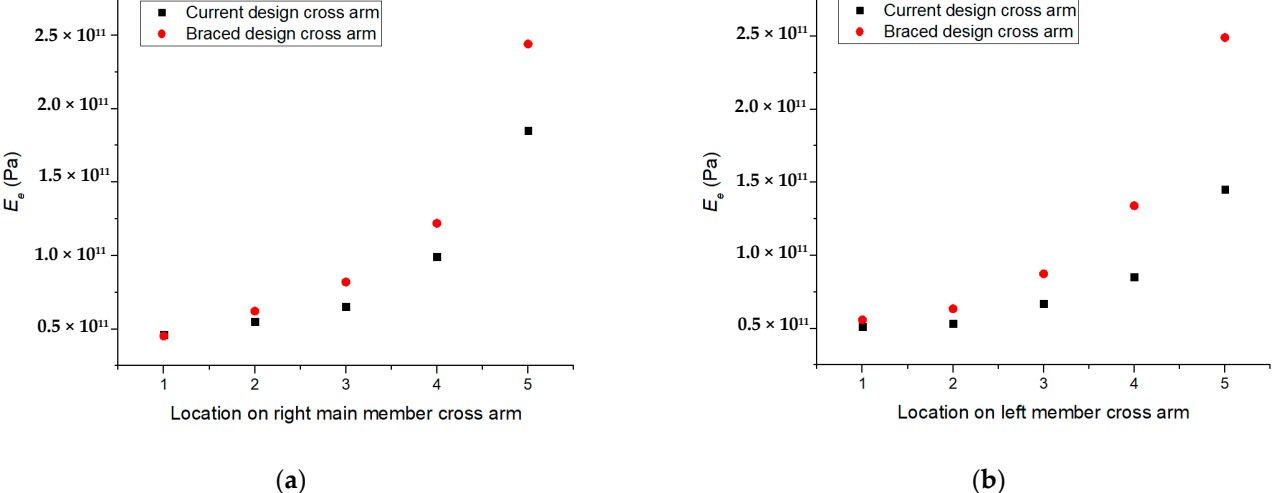

(**a**)                                        (**b**)

**Figure 11.** $E_e$ parameter for current and braced wooden cross-arms: (**a**) right; (**b**) left main members.

In this study, another parameter was being evaluated, $\eta_k$, which represented the relaxation coefficient for the viscoelastic property. Moreover, the viscoelastic parameter, also known as irrecoverable creep strain, demonstrated the relaxation response over time [23]. Figure 12 displays the viscoelastic properties for both cross-arms (braced and existing cross-arms), showing relatively the same values of viscoelastic modulus. This illustrated that the bracing system did improve the viscoelastic properties of the cross-arm's structure in terms of relaxation time especially at y5 location. This showed that the bracing arms increased the relaxation of the cross-arm under long-term constant load. Apart from that, both cross-arms exhibited linear viscoelastic property along the beam length since the set working load was below the critical value of applied stress. If a critical stress was achieved during the creep cycle, the creep rate grew disproportionately faster [50].

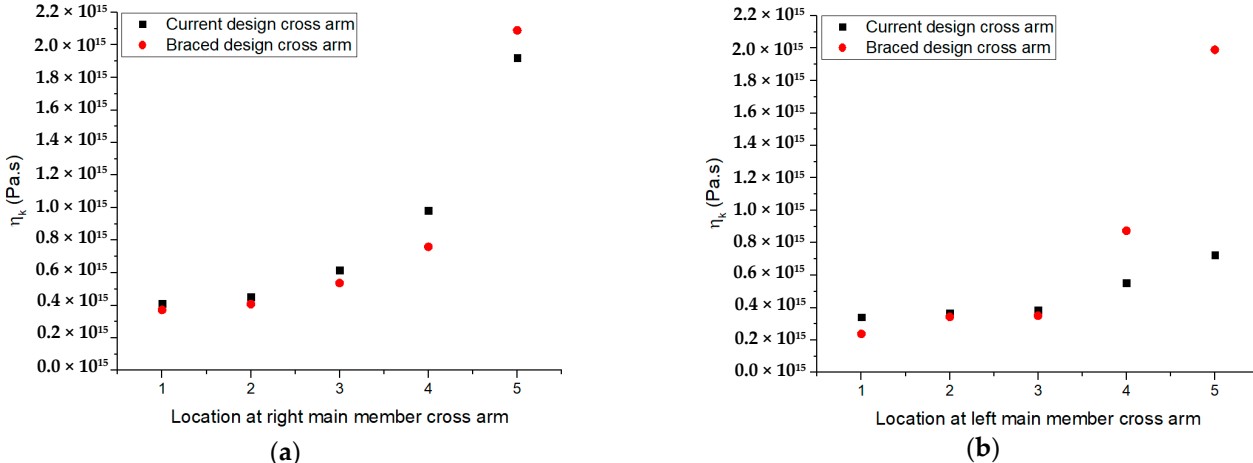

**Figure 12.** $\eta_k$ parameter for current and braced wooden cross-arms: (**a**) right; (**b**) left main members.

### 3.4. Creep Models Accuracy and Validation

In terms of accuracy of data plotted in creep strain-time analysis, the adjusted regression (Adj. $R^2$) or coefficient of determination was used to fit these numerical models with experimental outputs. These Adj. $R^2$ values are tabulated in Tables 2 and 3 for the Findley and Burger models, respectively. From these tables, it was found that the adj. $R^2$ for Findley's power law model exhibited higher values, ranging from 0.833 to 0.989 rather than the Burger model, ranging from 0.518 to 0.913. The Burger model forecasted a relationship of viscosity and time linearly, which was affected during the numerical model fitting with experimental data [51]. Thus, from the adjusted regression evaluation, the best numerical model suitable to analyse the finding of creep for wooden cross-arm was Findley's power law model. This observation showed that the wooden cross-arm experienced a steady-state creep in a long-term period and did not permit any sign of tertiary creep phase to suddenly fail [52,53] during long-term period and not permit any sign of tertiary creep phase to be suddenly failed. However, the project also required the use of the Burger model in order to examine the effect of addition of bracing arms on elastic and viscoelastic properties of wooden cross-arm. Thus, both studies were required to achieve a holistic view and analysis on creep behaviours for the existing and braced wooden cross-arms.

These creep models (Findley's power law model and Burger model) were validated by comparing their instantaneous strain with experimental results. Fundamentally, the instantaneous strains were principally proportional to the applied stress according to Hooke's law. Table 5 summarises the evaluation of instantaneous strain between the experimental outcomes with two numerical models (Findley and Burger models) for both existing and braced cross-arms at y3 location. Since the y3 location exhibited the most highly severe in terms of creep strain, thus, the creep strain results were compared with the numerical outputs. As seen in Table 5, all percentage errors recorded were below than 5%. This showed that all numerical models including the Findley and Burger models fitted the experimental data accurately. According to Zhang et al. [54], the acceptable value for percentage error when comparing the experimental outputs with numerical values should be less than 20%. When the percentage error below 20%, it displayed that the plotted experimental data were not severely deviated, and consistent within the principles proposed from the exact numerical model. In this study, the experimental data plotted to elaborate the creep properties of the wooden cross-arms were verified with precise and consistent values.

**Table 5.** Comparison of instantaneous strain value between experimental outputs and numerical models located at y3 for current and braced wooden cross-arms.

| Configuration | Model | Inst. Strain | Located at y3 at Main Member | | | |
|---|---|---|---|---|---|---|
| | | | Right | Percentage Error (%) | Left | Percentage Error (%) |
| Current cross-arm | Experimental data | $\varepsilon\ (10^{-3})$ | 1.006 | - | 0.988 | - |
| | Findley model | $\varepsilon_o\ (10^{-3})$ | 1.010 | 0.398 | 0.994 | 0.604 |
| | Burger model | $\varepsilon_o\ (10^{-3})$ | 1.010 | 0.398 | 0.986 | 0.202 |
| Braced cross-arm | Experimental data | $\varepsilon\ (10^{-3})$ | 0.806 | - | 0.731 | - |
| | Findley model | $\varepsilon_o\ (10^{-3})$ | 0.798 | 0.993 | 0.722 | 1.231 |
| | Burger model | $\varepsilon_o\ (10^{-3})$ | 0.806 | 0.000 | 0.756 | 3.420 |

## 4. Conclusions

The creep properties of Balau wood timber cross-arms reinforced with additional braced arms was significantly reduced as compared to existing design wooden cross-arms. Thus, the implementation of bracing system in cross-arm structures display a good potential for application existing wooden cross-arms in latticed transmission tower. Many previous studies conducted creep experiment of wooden specimens in laboratory with controlled environment. This approach is typical for intended baseline characterizations. However, no study has been carried out on full-scale size of wooden cross-arm in actual environment of transmission tower. This study is narrowed to compare the braced and current design of wooden cross-arms with actual working load and environment conditions. The comparison depicts that the creep strain of the main member for braced wooden cross-arm had reduced about 15–21% compared to existing design of wooden cross-arm. In addition, the addition of braced arms in cross-arm structures would effectively enhance the stability of the viscoelastic stage, which would reduce the failure probability. Additional creep analyses were carried out using Findley and Burger models discovered that braced wooden cross-arm has greater elastic modulus. This indicates that incorporation of extra braced arms contribute better flexure property for overall structure. The braced systems increased the viscoelastic modulus of the cross-arm, thus enhancing relaxation during creep. Moreover, the results also displays the braced wooden cross-arm permit more stable stress independent material exponent between right and left main members. This shows that bracing system in cross-arm would induce better dimensional stability of the structure. As a conclusion, the implementation of additional braced arms would be highly advantageous during construction of latticed transmission tower which could prolong the cross-arm's service life.

It is suggested that the connection of the braced arms with cross-arm's members could possibly affect the overall long-term mechanical properties of the structure. In the case of timber structures, the flexibility of wood in connection, and the flexibility of the connectors themselves have a very significant impact on the deformation of the entire structure. This aspect is vital for the creep study of a complex structure in order to further elaborate the potential failures and problems in the future. Thus, it is highly suggested that further study could be conducted to examine the effect of braced arms connection on creep behaviours of the wooden cross-arm.

**Author Contributions:** Conceptualization, M.R.I. and M.R.M.A.; methodology, M.R.M.A.; software, M.R.M.A.; validation, M.R.I., S.M.S. and N.Y.; investigation, M.R.M.A.; resources, M.R.I.; writing—original draft preparation, M.R.M.A.; writing—review and editing, M.R.M.A., and M.R.I.; visualization, M.R.M.A.; supervision, M.R.I., S.M.S. and N.Y.; project administration, M.R.I.; funding acquisition, M.R.I. All authors have read and agreed to the published version of the manuscript.

**Funding:** This research work was funded by Ministry of Higher Education, Malaysia for financial support through the enumerator service under Fundamental Research Grant Scheme (FRGS): FRGS/1/2019/TK05/UPM/02/11 (5540205) and Geran Putra: (9634000) by Universiti Putra Malaysia to carry out all research activities.

**Institutional Review Board Statement:** Not applicable.

**Informed Consent Statement:** Not applicable.

**Data Availability Statement:** The data used to support the findings of this study are included within the article.

**Acknowledgments:** This research work was funded by Ministry of Higher Education, Malaysia for financial support through the enumerator service under Fundamental Research Grant Scheme (FRGS): FRGS/1/2019/TK05/UPM/02/11 (5540205) and Geran Putra: (9634000) by Universiti Putra Malaysia to carry out all research activities. The authors are also very thankful to Department of Aerospace Engineering, Faculty of Engineering, UPM for providing space and facilities for the project. Moreover, all authors are very appreciate and thankful to Jabatan Perkhidmatan Awam (JPA) and Kursi Rahmah Nawawi for providing scholarship award and financial aids to the first author to carry out this research project.

**Conflicts of Interest:** The authors declare no conflict of interest.

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
