# Peer review of "Influence of Additional Bracing Arms as Reinforcement Members in Wooden Timber Cross-Arms on Their Long-Term Creep Responses and Properties"

_applsci, doi:10.3390/app11052061_

Round 1

Reviewer 1 Report

The research problem posed is interesting and important from a practical point of view – the Authors mention the problems reported in relation to the realized structures of the analyzed type. For the construction of the timber high voltage tower, it was proposed to introduce a structural change, and an analysis of its effect on long-term creep was carried out. As part of the research, tests were carried out on a full-scale model and then compared with the results of numerical analyzes based on selected theoretical models.

However, the way the experiment is presented is insufficient and does not allow for a proper assessment of the correctness of the research on the model. In particular:

  • No information was given on how many samples the experiment was performed on. The authors themselves admit that the tested material is heterogeneous (Shorea Dipterocarpaceae wood). Relying on a single sample without statistical analysis of the variability of material characteristics reduces the reliability of the research and its possible practical usefulness.
  • The description of the testing stand is too simplified. The drawing lacks basic dimensions (information provided in the text is insufficient in this respect). There is no information about the testing rig rigidity.
  • The influence of connections between individual bars has been almost completely ignored. The text mentions only generally that they are used "both bolts and nuts, as well as mild steel fastening brackets". In the case of timber structures, the flexibility of wood in connection, and the flexibility of the connectors themselves have a very significant impact on the deformation of the entire structure.

The numerical analysis seems correct. However, the final conclusions do not take into account the impact of a possible change in the amount of stress on the creep. Authors should consider supplementing the paper before publication.

Author Response

The research problem posed is interesting and important from a practical point of view – the Authors mention the problems reported in relation to the realized structures of the analyzed type. For the construction of the timber high voltage tower, it was proposed to introduce a structural change, and an analysis of its effect on long-term creep was carried out. As part of the research, tests were carried out on a full-scale model and then compared with the results of numerical analyzes based on selected theoretical models.

However, the way the experiment is presented is insufficient and does not allow for a proper assessment of the correctness of the research on the model. In particular:

Point 1: No information was given on how many samples the experiment was performed on. The authors themselves admit that the tested material is heterogeneous (Shorea Dipterocarpaceae wood). Relying on a single sample without statistical analysis of the variability of material characteristics reduces the reliability of the research and its possible practical usefulness. 

Response 1: Thank you for your suggestion. However, the wooden timber cross arm used in experiment has same geometry and dimensions (specifications) as used by Tenaga Nasional Berhad (TNB), the only power supply company in Peninsular Malaysia. The source of wooden cross arm is limited in supply as most of the stocks are produced based on TNB’s order. Therefore, the specimen used in the experiment was only after getting authorisation and approval by TNB, which limited in the unit and controlled by them. This condition resulted in the lack of replication in the experiment. This work also is at the exploration stage of the research. 

Point 2: The description of the testing stand is too simplified. The drawing lacks basic dimensions (information provided in the text is insufficient in this respect). There is no information about the testing rig rigidity.

Response 2: Thank you for your comments and suggestions. The description of the creep test rig has been further elaborated in Lines 108-110. The basic dimension of creep test rig has been added in Figure 2. The information of testing rig rigidity has been added in Table 1.

Point 3: The influence of connections between individual bars has been almost completely ignored. The text mentions only generally that they are used "both bolts and nuts, as well as mild steel fastening brackets". In the case of timber structures, the flexibility of wood in connection, and the flexibility of the connectors themselves have a very significant impact on the deformation of the entire structure.

Response 3: Thank you for your suggestions. The influence of connection between individual bars has been summarized and discussed in Lines 365-371.

Point 4: The numerical analysis seems correct. However, the final conclusions do not take into account the impact of a possible change in the amount of stress on the creep. Authors should consider supplementing the paper before publication.

Response 4: Thank you for your comment and appreciate it. The authors has altered and discussed the Conclusion section in Lines 345-364. The data used to support the findings of this study are included within the article such as Table 1 and 3.

Reviewer 2 Report

The paper presents a study related to the cross arms used in transmission tower: the possibility to retrofit the bracing arms is investigated and in particular issue about creep phenomena are discussed.

Cross arms components consist of three members (two main members and one tie member: the aforementioned system was tested in order to simulate the creep in actual environment of tropical climate conditions. Tests results have been discussed with two numerical models.

Authors demonstrates that the addition of bracing arms improve the dimensional and structure stability in the respect of the long-term mechanical properties. 

For a sake of comprehension, it would be useful have some information regarding the transmission tower (dimensions, geometry) and in particular regarding the connection between the steel tower to the timber arms.

For example, in relation to Figure 3. It is recommended to implement the figure or add a new figure where the detail of the connection is represented. Please report and describe the used connection and how the bracing members are connected to the main members (it is not simple to understand the 3D disposition of the timber elements).

Figure 5: please respect the size of other figures. This figure seems to be too large.

Figure 6. please explain how is calculated the creep in particular it is not clear why it is expressed as (mm/mm)

In the reviewer opinion it is important to considerate the role of the connection in the long term deformation: the paper studied the creep phenomena of the timber member (this aspect is properly studied and described) but, since the problem of the long term deformation of the connections, was not discussed, at least some sentences to introduce this aspect are required.

Please rephrase the conclusion, some sentences are not clear.

Author Response

The paper presents a study related to the cross arms used in transmission tower: the possibility to retrofit the bracing arms is investigated and in particular issue about creep phenomena are discussed.

Cross arms components consist of three members (two main members and one tie member: the aforementioned system was tested in order to simulate the creep in actual environment of tropical climate conditions. Tests results have been discussed with two numerical models.

Authors demonstrates that the addition of bracing arms improve the dimensional and structure stability in the respect of the long-term mechanical properties.

Point 1: For a sake of comprehension, it would be useful have some information regarding the transmission tower (dimensions, geometry) and in particular regarding the connection between the steel tower to the timber arms. 

Response 1: Thank you for your suggestion. The information of the transmission tower such as geometry and dimension have been added at Lines 110-113 and Figure 3. The details of connection between steel tower and timber arms has been added in Lines 120-126 and Figure 4.

Point 2: For example, in relation to Figure 3. It is recommended to implement the figure or add a new figure where the detail of the connection is represented. Please report and describe the used connection and how the bracing members are connected to the main members (it is not simple to understand the 3D disposition of the timber elements).

Response 2: Thank you for your suggestions. The details and elaborations of the connections has been added in Lines 127-135 and Figure 5.

Point 3: Figure 5: please respect the size of other figures. This figure seems to be too large.

Response 3: Thank you for your comments. The authors have overlooked. Thus, we have resized the figure accordingly.

Point 4: Figure 6. please explain how is calculated the creep in particular it is not clear why it is expressed as (mm/mm).

Response 4: Thank you for your comments. The authors have overlooked. It is supposedly to write in % as the strain is equal to stress divided by modulus. We have corrected the units as in Figure 9.

Point 5: In the reviewer opinion it is important to considerate the role of the connection in the long term deformation: the paper studied the creep phenomena of the timber member (this aspect is properly studied and described) but, since the problem of the long term deformation of the connections, was not discussed, at least some sentences to introduce this aspect are required.

Response 5: Thank you for your comments. The role of connection in the long term deformation of cross arm is suggested and elaborated in Lines 365-371.

Point 6: Please rephrase the conclusion, some sentences are not clear

Response 6: Thank you for your comments. The conclusion sections has been altered and added with proper discussion in Lines 345-364.
